# Non-Pharmacological Management of Hypertension: Exploring Determinants for Optimizing Physical Activity Implementation in Cameroon

**DOI:** 10.3390/ijerph23010051

**Published:** 2025-12-31

**Authors:** Maurice Douryang, Hyacinte Trésor Ghassi, Dilane Landry Nsangou Muntessu, Steve Ulrich Endeksobo, Borel Idris Djike Noumsi, Annick Cindy Fah Nono Mefo, Leonard Tanko Tankeng, Florian Forelli

**Affiliations:** 1Department of Physiotherapy and Physical Medicine, University of Dschang, Dschang 96, Cameroon; 2Doctoral School, Evangelical University Institute of Cameroon, Bandjoun 127, Cameroon; 3Department of Physiotherapy, Centre Polyvalent de Formation de Mbouo-Bandjoun, Bandjoun, Cameroon; 4Department of Surgery and Specialities, University of Douala, Douala 2701, Cameroon; 5Haute-Ecole Arc Santé, HES-SO University of Applied Sciences and Arts Werstern Switzerland, 2800 Delémont, Switzerland; 6Orthopaedic Surgery Department, Clinic of Domont, Ramsay Healthcare, Ortholab, 95330 Domont, France; 7SFMKS Lab, 95270 Asnières sur Oise, France

**Keywords:** hypertension, physical activity, determinants, barriers, Cameroon, noncommunicable diseases

## Abstract

**Highlights:**

**Public health relevance—How does this work relate to a public health issue?**
Physical inactivity remains highly prevalent among hypertensive patients in Cameroon.Key personal and environmental determinants influence adherence to WHO physical activity recommendations.

**Public health significance—Why is this work of significance to public health?**
Identifying sociodemographic and clinical predictors of inactivity helps target high-risk groups.Understanding locally reported barriers and facilitators guides more culturally adapted interventions.

**Public health implications—What are the key implications or messages for practitioners, policy makers and/or researchers in public health?**
Integrating structured physical-activity counselling into routine hypertension care is urgently needed.Policymakers should strengthen community-level PA support, especially for older adults, women, and patients with chronic conditions.

**Abstract:**

Physical activity (PA) is a cornerstone of non-pharmacological hypertension management, yet evidence on its determinants remains limited in African populations. We conducted a cross-sectional study among 383 hypertensive patients in two referral hospitals in Cameroon to assess PA levels and associated factors. PA was classified as insufficiently active (<600 MET-min/week) or active (≥600 MET-min/week). Overall, 54% of participants were insufficiently active, 37.9% had moderate activity, and 8.1% reported vigorous activity. Older age was strongly associated with inactivity, particularly for ages 60–74 (aOR = 2.84, *p* < 0.001) and for ≥75 years (aOR = 18.67, *p* < 0.001). Comorbidities also predicted inactivity, including renal failure (aOR = 2.41, *p* < 0.001) and diabetes/other complaints (aOR = 4.92, *p* < 0.001). Female sex increased the odds of inactivity (aOR = 1.42, *p* = 0.038). Whereas higher education was protective, particularly secondary (aOR = 0.12, *p* < 0.001) and high-school level (aOR = 0.05, *p* < 0.001). Among inactive participants, the most frequent barriers were lack of motivation (38.6%), physical impairment (37.2%), lack of prescription (23.2%), and space constraints (21.7%), whereas perceived benefits (39.1%), motivation (26.1%), and available space (32.4%) were the most cited facilitators; however, none of these factors showed a significant association with PA in chi square analysis. The high prevalence of inactivity and the strong influence of sociodemographic and clinical characteristics underscore the need for tailored interventions that target older adults, women, and patients with comorbidities, while strengthening education and structured support for PA within hypertension care pathways.

## 1. Introduction

Hypertension remains one of the most prevalent chronic diseases worldwide, affecting nearly 1 billion individuals and accounting for about 7.1 million deaths each year [1]. It represents a major contributor to global mortality, affecting approximately 40% of adults and significantly increasing the risk of stroke and cardiovascular disease [2,3]. Globally, up to 10 million deaths annually can be attributed to arterial hypertension [4].

According to the World Health Organization (WHO), quality of life is defined as an individual’s perception of their position in life within the cultural and value systems in which they live, and in relation to their goals, expectations, standards, and concerns [5]. A good quality of life contributes not only to individual well-being but also to sustainable societal development [6]. Among hypertensive individuals, quality of life is influenced by several clinical and psychosocial factors, including comorbidities, blood pressure control, physical activity level, salt intake, body mass index (BMI), hypertension grade, treatment adherence, family support, autonomy, and stress level [7]. In Sub-Saharan Africa and most low-resource countries, the prevalence of hypertension continues to rise, primarily due to increasing sedentary behaviour, poor dietary habits, and psychosocial stress [8]. In Cameroon, recent findings published in 2019 revealed a quite high prevalence rate, indicating that nearly four out of ten adults are affected by hypertension [9].

Lifestyle modification remains the first-line therapeutic strategy for managing high blood pressure, with physical activity (PA) being a cornerstone of non-pharmacological intervention [3]. Adherence to healthy lifestyle behaviours such as maintaining a normal BMI and waist circumference, engaging in regular PA, avoiding smoking, moderating alcohol intake, following the DASH diet, and consuming supplements like garlic, cocoa, vitamin C, coenzyme Q10, omega-3 fatty acids, calcium, potassium, and magnesium has been shown to improve cardiovascular outcomes and reduce blood pressure in hypertensive patients [10,11,12,13].

The European Society of Cardiology recommends that individuals with hypertension engage in moderate-intensity aerobic physical activity (such as walking, jogging, cycling, or swimming) for at least 30 min per day, five to seven days per week, to prevent and manage hypertension and reduce the risk of cardiovascular morbidity and mortality [14]. Nevertheless, insufficient physical activity remains a major public health concern worldwide and contributes to an increased risk of chronic noncommunicable diseases [15].

Evidence from a systematic review conducted in 18 low- and middle-income countries demonstrated that physical activity interventions can lower systolic blood pressure by an average of 8 mmHg and diastolic pressure by 4 mmHg [16]. Despite these well-documented benefits, physical activity is still not routinely prescribed as a therapeutic approach in the management of hypertension in Cameroon. Consequently, no study to date has examined the physical activity levels of hypertensive patients or identified the factors influencing their adherence to regular exercise. This lack of evidence limits the implementation of effective strategies for promoting and supporting physical activity among hypertensive individuals. Therefore, the present study aimed to assess the physical activity practices of hypertensive patients in Cameroon and to identify the facilitators and barriers influencing adherence to regular physical activity.

## 2. Materials and Methods

### 2.1. Study Design, Setting, and Period

This cross-sectional analytical study was conducted between 1 June and 5 November 2025, with data collection occurring from 1 September to 30 October 2025, in the West region of Cameroon. Participants were recruited from the Regional Hospital of Bafoussam and the Mifi District Hospital, two major referral hospitals in the region that receive patients from several surrounding areas, including the Northwest, Southwest, Centre, Littoral, and East regions of Cameroon.

### 2.2. Study Population

The study population comprised patients with a confirmed diagnosis of hypertension (all types) receiving follow-up care in the selected facilities. Inclusion criteria were: a documented diagnosis of hypertension established by a licenced healthcare professional and recorded in the patient’s medical file, and provision of written informed consent. Patients with incomplete data or missing interview responses were excluded from the analysis. Participants were selected using a consecutive sampling approach, whereby all hypertensive patients attending routine follow-up consultations during the data collection period and meeting the inclusion criteria were invited to participate.

### 2.3. Sample Size Determination

The primary study endpoint was physical activity (PA) practice. The minimum required sample size for estimating a single proportion was determined using the cochrane formula Assuming a Z-value of 1.96, an anticipated prevalence (*p*) of 0.50 (a conservative estimate in the absence of prior data in Cameroon), and a precision (d) of ±10% at a 95% confidence level, the calculated minimum sample size was 384 participants.

### 2.4. Data Collection and Instrument

After obtaining ethical clearance and institutional authorizations, data were collected through face-to-face interviews conducted by trained physiotherapists using a structured questionnaire. The tool captured three main domains: Sociodemographic and clinical characteristics: age, sex, educational level, duration since hypertension diagnosis, Body weight and height were measured by trained physiotherapists using standardized equipment, and body mass index (BMI) was calculated from these objective measurements; we also assessed the healthcare provider responsible for follow-up, adherence to prescribed medication, use of traditional medicine, and presence of comorbidities. Physical activity level assessed using the International Physical Activity Questionnaire–Short Form (IPAQ-SF). This validated instrument classifies activity into low, moderate, and vigorous intensity categories, expressed in metabolic equivalent task (MET)-minutes per week. Participants were categorized as: Inactive: <600 MET-minutes/week (Low activity), Active: ≥600 MET-minutes/week (moderate to high activity). The IPAQ-SF has demonstrated strong reliability and cross-cultural applicability in assessing PA in epidemiological studies [17]. Perceived barriers and facilitators to PA: including self-reported barriers (e.g., lack of time, motivation, financial means, medical prescription, family support, or available spaces for PA, and physical limitations) and facilitators (e.g., knowledge of PA benefits, access to suitable environments, family encouragement, and personal motivation). The list of perceived barriers and facilitators was developed by the research team based on items commonly used in PA studies in Africa and LMICs, WHO STEPS and IPAQ-related guidance, and themes from local physiotherapy practice. The final items were reviewed by two senior physiotherapists and a public health expert to ensure cultural relevance and content validity.

### 2.5. Data Quality Assurance and Bias Minimization

To ensure data validity and reliability: Physiotherapists underwent standardized training before data collection. Information on diagnosis and disease duration was verified against medical records to reduce recall bias. The questionnaire was pretested on a small sample for clarity. Completed questionnaires were cross-checked by two independent investigators, and all entries were double-entered into the database to minimize transcription errors.

### 2.6. Statistical Analysis

Data analysis was performed using IBM SPSS Statistics version 23. Continuous variables were summarized as means ± standard deviations (SD), while categorical variables were expressed as frequencies and percentages. The Chi-square test (χ^2^) was applied to explore associations between potential barriers/facilitators and PA practice. Then factors significantly associated with PA were introduced in multivariable logistic regression, where crude and adjusted odds ratios were calculated. The level of significance was set at *p* < 0.05, with a 95% confidence interval (CI). Although the IPAQ-SF classifies physical activity into three categories (low, moderate, and vigorous), we dichotomized PA levels using the <600 versus ≥600 MET-min/week threshold in order to identify participants who did or did not meet the minimum WHO recommendations for cardiovascular health. In this study, individuals with <600 MET-min/week were considered inactive (i.e., insufficiently active to achieve health benefits), whereas those accumulating ≥600 MET-min/week were categorized as active, encompassing both moderate and vigorous activity levels. This cut-off is widely used in epidemiological studies evaluating adherence to PA guidelines among hypertensive populations [17].

### 2.7. Ethical Considerations

Ethical approval was obtained from the Ethics and Human Health Research Committee of the Western Region of Cameroon (Ref: 823/27/08/2025/CE/CRERSH-OU/VP). Written informed consent was obtained from all participants. All data were anonymized, coded, and handled confidentially, and interviews were conducted in private settings to ensure participant privacy.

### 2.8. Reporting Guidelines

This study followed the Strengthening the Reporting of Observational Studies in Epidemiology (STROBE) guidelines for cross-sectional studies.

## 3. Results

### 3.1. General Characteristics of Participants

From the 400 patients investigated, we excluded 17 due to missing data. Were finally included in this study 383 participants. Among participants, the average duration of hypertension was 4.6 ± 3.7 years, and the mean Body Mass Index (BMI) was 27.4 ± 3.7 kg/m^2^. Table 1 presents the distribution of participants’ sociodemographic characteristics. There was a balanced representation across age groups, with 32.1% aged 60–74, 21.9% aged 45–59, 18% aged 30–44, and 9.9% aged 20–29. The male-to-female sex ratio was 1.6. Regarding education, 37.9% of participants had a secondary level and 30.8% had a university-level education. Most participants were followed-up by medical doctors (59.5%), followed by cardiologists (27.7%).

### 3.2. Treatment and Clinical Status of Participants

Table 2 presents key indicators related to treatment and comorbidities. All participants reported receiving medications (100%), while only 1% reported using additional traditional treatments. Regarding other diseases, nearly half of the respondents (48.6%) reported no additional conditions. However, renal failure and diabetes were each reported by 25.3%. Conditions such as stomach pain, cough, and osteoarthritis were rare (0.3%) each.

### 3.3. Self-Reported Barriers and Facilitators of Physical Activity and Physical Activity Levels

According to the levels of physical activity, 207 (54%) participants reported low physical activity practice, 145 (37.9%) reported moderate physical activity practice while only 31 (8.2%) were engaged in intense physical activity. Therefore, according to that, 54% of participants where inactive, while 46% were active. Table 3 highlights the main barriers and facilitators to engaging in physical activity. Among inactive participants, the most frequently reported barriers were lack of motivation (38.6%) and physical impairment (37.2%), followed by lack of prescription (23.2%), lack of support (21.3%), and space constraints (21.7%). Active individuals reported a similar pattern, with lack of motivation (36.4%) and physical impairment (34.1%) also appearing most often.

Regarding facilitators, perceived benefits were the most cited among inactive participants (39.1%), followed by motivation (26.1%) and available space (32.4%). Active individuals reported comparable proportions, with available space (30.7%), motivation (22.4%), and perceived benefits (38.6%) being the most frequent.

### 3.4. Factors Associated with Physical Activity Practice Among Participants

Table 4 shows that age, sex, education level, and comorbidities were significantly associated with physical activity status, while perceived barriers and facilitators were not. Age demonstrated the strongest gradient effect (χ^2^ = 85.8; *p* < 0.0001), with inactivity increasing substantially in older groups. Sex was also significantly associated with activity level (χ^2^ = 5.27; *p* = 0.022), with women more likely to be inactive. Educational level showed a clear inverse relationship with inactivity (χ^2^ = 72.4; *p* < 0.0001), indicating that higher education is linked to greater activity engagement. Comorbidities were also significantly associated with inactivity (χ^2^ = 54.01; *p* < 0.0001). In contrast, perceived barriers (χ^2^ = 1.63; *p* = 0.90) and facilitators (χ^2^ = 0.06; *p* = 0.996) did not show significant associations. Based on these findings, variables that were significantly associated in bivariate analysis were entered into the multivariable logistic regression model.

In the multivariable logistic regression model, several factors remained independently associated with physical inactivity among hypertensive patients. Age showed the strongest effect: compared with adults aged 20–29 years, the odds of inactivity increased progressively with age, reaching the highest risk among those aged 75 years and older (aOR = 18.67, *p* < 0.001). Female participants also had significantly higher odds of being inactive than males (aOR = 1.42, *p* = 0.038). Educational attainment demonstrated a robust protective effect, with primary (aOR = 0.31, *p* < 0.001), secondary (aOR = 0.12, *p* < 0.001), and high-school education (aOR = 0.05, *p* < 0.001) all associated with substantially lower odds of inactivity compared with no education. Comorbid conditions were also strong predictors of inactivity: participants with renal failure (aOR = 2.41, *p* < 0.001) or diabetes/other complaints (aOR = 4.92, *p* < 0.001) had significantly higher odds of inactivity than those without additional diseases. Together, these findings indicate that sociodemographic and clinical factors rather than perceived barriers or facilitators play a central role in determining physical activity engagement in this population (Table 5).

## 4. Discussion

This study investigated physical activity (PA) levels and their determinants among hypertensive patients in two referral hospitals in Cameroon. The findings revealed that more than half of participants (54%) reported insufficient physical activity, while only 8.1% engaged in intense activity. This low participation rate highlights a major gap in the non-pharmacological management of hypertension, despite universal access to antihypertensive medication in this population. This low rate of participation in PA could be explained by the lack of prescription and encouragement of PA by clinicians following hypertensive patients, as this absence of prescription was identified as a high self-reported barrier to PA practice among participants. However, this inactivity rate is similar to other studies in the literature [18,19,20]. This suggests that the integration of PA as a non-pharmacological treatment remains limited among hypertensive patients, despite its widely proven effects [21,22,23]. Therefore, these results suggest that strategies should be implemented to promote PA as an integral component of hypertension management in Cameroon and similar settings. Clinicians, as central actors in this process, should systematically prescribe and encourage PA alongside pharmacological treatment.

Sociodemographic factors, including age, sex, and educational level, were significantly associated with PA engagement. Higher participation among men is consistent with findings from studies in Nigeria [24], Ghana [25], where women reported lower levels of physical activity due to domestic responsibilities, limited autonomy, and cultural norms that discourage vigorous exercise. Additionally, the positive association between educational attainment and PA levels aligns with evidence from South Africa [26,27] and Kenya [28,29], where individuals with higher education are more aware of the health benefits of PA and have better access to facilities and information.

Older age was strongly associated with physical inactivity in our sample, with the odds of being inactive rising steeply from midlife onwards and remaining highest among participants aged ≥ 75 years. This age-related gradient is consistent with recent evidence from hypertensive cohorts and general adult populations in low- and middle-income countries, where older adults are less likely to achieve WHO physical activity recommendations than younger adults. For example, Belay et al. reported that older hypertensive patients in Northwest Ethiopia had markedly higher odds of inadequate physical activity compared with younger patients, highlighting age as a key barrier to guideline-concordant activity in this clinical group [30]. Similarly, in rural Kenya, Bosire et al. found that adults aged ≥ 65 years were over twice as likely to be inactive as younger adults, underlining how ageing interacts with environmental and social constraints to limit engagement in regular physical activity [28]. Large multi-country surveillance data from African STEPS surveys also show that older adults have persistently lower levels of leisure-time physical activity than younger age groups [31]. Taken together, these findings support the need for age-adapted and function-tailored activity programmes for older adults with hypertension.

We also observed that women were more likely to be physically inactive than men, even after adjustment for sociodemographic and clinical factors. This sex disparity aligns with recent data in hypertensive populations and in the wider African context. In rural Kenya, Bosire et al. reported that women had nearly twice the odds of being physically inactive compared with men, despite living in the same communities and sharing similar environmental constraints [28]. Among more than 7000 adults with hypertension in Korea, Jung et al. showed that women were over-represented in the “no physical activity” category and under-represented in the ideal physical activity group compared with men [32]. At the continental level, Oyeyemi et al. recently demonstrated that women across 11 African countries consistently report lower levels of leisure-time physical activity than men, despite sometimes higher participation in transport or occupational domains [31]. Our findings therefore reinforce the need for gender-sensitive interventions that address structural, cultural, and caregiving barriers that disproportionately limit women’s ability to engage in regular physical activity.

Comorbid chronic conditions such as renal failure and diabetes were independently associated with higher odds of physical inactivity in our hypertensive population. This pattern is in line with emerging evidence on multimorbidity and movement behaviour. In a recent accelerometer-based study from the Danish Lolland-Falster Health Study, Jørgensen et al. showed that adults with multimorbidity spent almost half of their waking day sedentary and that the proportion not meeting WHO physical activity guidelines increased from 63% in those with two conditions to more than 80% in those with four or more conditions [33]. The same study reported an inverse dose–response relationship between the number of chronic conditions and daily step counts, suggesting that physical inactivity intensifies as disease burden becomes more complex [33]. Among hypertensive patients in Ethiopia, Belay et al. similarly found that poor self-rated health and limited access to facilities were associated with inadequate physical activity [30]. Our results extend this literature by showing that specific cardiometabolic comorbidities (renal failure and diabetes/other complaints) are strong correlates of inactivity in hypertensive patients, underscoring the importance of integrated, multimorbidity-aware exercise counselling in routine hypertension care.

Finally, higher educational attainment emerged as a robust protective factor against physical inactivity, with secondary and high-school education associated with substantially lower odds of being inactive. This is consistent with recent African and global data showing that adults with higher education are more likely to meet physical activity recommendations. In a large analysis of secular trends in 11 African countries, Oyeyemi et al. reported that individuals with lower education consistently had lower levels of leisure-time physical activity than their more educated counterparts [31]. In rural Kenya, Bosire et al. also identified socio-cognitive factors linked to education, such as health literacy and exposure to health information as important correlates of physical activity behaviour [28]. These findings support our interpretation that education may enhance awareness of the benefits of physical activity, improve navigation of health services, and increase access to safe spaces or structured programmes for exercise. Targeting individuals with lower education through tailored counselling and community-based programmes may therefore be critical for reducing physical inactivity among hypertensive patients.

## 5. Conclusions

This study shows that physical inactivity remains highly prevalent among hypertensive patients in Cameroon and is strongly influenced by age, sex, education, and comorbidities. While perceived barriers and facilitators were frequently reported, they were not independently associated with activity status after adjustment, highlighting the predominance of sociodemographic and clinical determinants in shaping behaviour. These findings underscore the need to integrate structured physical-activity counselling into routine hypertension care, strengthen patient education, and develop supportive environments that encourage safe and sustainable activity. Implementing such strategies may enhance adherence to non-pharmacological management and ultimately improve cardiovascular outcomes in hypertensive populations in Cameroon and similar settings.

### 5.1. Clinical and Policy Implementation

Given our findings, notably the higher prevalence of physical inactivity among older participants, women, and those with comorbidities, health-care providers and policy-makers in Cameroon should consider integrating structured physical-activity support into routine hypertension care. For example, including physiotherapists, community health workers, or trained exercise counsellors in hypertension clinics could help to design safe, tailored exercise programmes adapted to patients’ comorbid conditions and age.

At the community and primary-care level, educational initiatives should be strengthened. Health-education campaigns (through clinics or community gatherings) could raise awareness about the benefits of regular physical activity, especially targeting individuals with lower educational attainment, who may lack information or motivation. Supportive interventions might include safe, accessible environments for walking or low-intensity exercise, as well as group sessions to encourage peer support and overcome barriers such as “lack of motivation” or “physical impairment.”

Finally, health-system policies should promote non-pharmacological management of hypertension by incorporating regular physical-activity counselling and follow-up in standard treatment protocols. This could involve training clinicians to assess physical activity levels, educating patients about WHO PA recommendations, and encouraging a multidisciplinary approach (e.g., doctors + physiotherapists + counsellors) to foster lifestyle changes alongside pharmacological treatment.

### 5.2. Limitations

This study present certain limitations that should be acknowledged. The cross-sectional design prevents inference of causal relationships between determinants and physical activity. Self-reported responses on activity levels and perceptions may be influenced by recall and social desirability biases, despite quality assurance measures. The hospital-based sampling limits representativeness, as patients attending referral hospitals may differ from those managed in community settings. Additionally, unmeasured factors such as income level, occupational workload, or neighbourhood safety might confound observed associations. We acknowledge that internal motivation and self-efficacy are conceptually distinct constructs. Also, as IPAQ-SF, perceived-barriers and facilitators items rely on self-report, responses may be affected by recall or social desirability bias. Although our questionnaire assessed motivation, it did not incorporate a validated self-efficacy scale. This represents a methodological limitation, as self-efficacy is recognized as one of the strongest predictors of physical activity behaviour. Future studies should include standardized measures of self-efficacy to better capture its influence on physical activity adherence among hypertensive patients. Hypertension grade could not be reported because this information was not consistently documented in participants’ medical records. As a result, hypertension severity was not included in the analyses, which may limit the clinical contextualization of the findings. Data were collected over a restricted period, which may not account for seasonal fluctuations in physical activity. Despite these constraints, the inclusion of participants from diverse regions of Cameroon enhances the external validity of findings. The results are therefore reasonably generalizable to hypertensive populations in similar Sub-Saharan African contexts, providing valuable insights to guide targeted health promotion interventions.

## Figures and Tables

**Table 1 ijerph-23-00051-t001:** Sociodemographic characteristics.

Variables	Frequency (*n* = 383)	Percentage (%)
Age (years)		
20–29	38	9.9
30–44	69	18
45–59	84	21.9
60–74	123	32.1
75 years and above	69	18
Sex		
Male	233	60.8
Female	150	39.2
Level of Education		
None	55	14.4
Primary	65	17
Secondary	145	37.9
High school	118	30.8
Professionnal following patients		
Medical doctor	228	59.5
Physiotherapist	16	4.2
Neurologist	32	8.4
Cardiologist	107	27.9

**Table 2 ijerph-23-00051-t002:** Types of medications and associated diseases.

Variables	Frequency (*n* = 383)	Percentage (%)
Taking medical drugs		
Yes	383	100
No	0	0
Traditional treatment		
Yes	4	1
No	379	99
Other diseases		
None	186	48.6
Renal failure	97	25.3
Diabetes	97	25.3
Stomach pain	1	0.3
Cough	1	0.3
Osteoarthritis	1	0.3

**Table 3 ijerph-23-00051-t003:** Barriers and facilitators of physical activity and Physical Activity levels.

Variables	Level of Physical Activity	
	Inactive *n* = 207 (%)	Active *n* = 176 (%)
Barriers		
Lack of time	17 (8.2)	18 (10.2)
Lack of motivation	80 (38.6)	64 (36.4)
Financial constraints	3 (1.4)	2 (1.1)
No prescription	48 (23.2)	35 (19.9)
Lack of support	44 (21.3)	40 (22.7)
Space constraints	45 (21.7)	39 (22.2)
Physical impairment	77 (37.2)	60 (34.1)
Facilitators		
Perceived benefits	81 (39.1)	68 (38.6)
Family support	54 (26.1)	47 (26.7)
Available space	64 (30.9)	54 (30.7)
Motivation	67 (32.4)	57 (32.4)

**Table 4 ijerph-23-00051-t004:** Factors associated with physical activity after Chi-square test.

Variables	Physical Activity			
	Frequency (%)		Chi Square	*p* Value
	**Inactive**	**Active**		
Age				
20–29	9 (4.3)	29 (16.5)	**85.8**	**<0.0001**
30–44	16 (7.7)	53 (30.1)		
45–59	48 (23.2)	36 (20.5)		
60–74	69 (33.3)	54 (30.7)		
75 years and above	65 (31.4)	4 (2.3)		
Sex				
Male	115 (55.6)	118 (67.0)	**5.27**	**0.022**
Female	92 (44.4)	58 (33.0)		
Level of Education				
None	51 (24.6)	4 (2.3)	**72.4**	**<0.0001**
Primary	48 (23.2)	17 (9.7)		
Secondary	73 (35.3)	72 (40.9)		
High school	35 (16.9)	83 (47.2)		
Professional following patients				
Medical doctor	113 (54.6)	115 (65.3)	2.30	0.51
Physiotherapist	11 (5.3)	5 (2.8)		
Neurologist	20 (9.7)	12 (6.8)		
Cardiologist	63 (30.4)	44 (25.0)		
Other diseases				
None	66 (31.9)	120 (68.2)	**54.01**	**<0.0001**
Renal failure	62 (30.0)	35 (19.9)		
Diabetes/other complaints	79 (38.2)	21 (11.9)		
Barriers				
Lack of time	17 (5.4)	18 (7.0)	1.63	0.90
Lack of motivation	80 (25.5)	64 (24.8)		
No prescription/Financial constraints	51 (16.2)	37 (14.3)		
Lack of support	44 (14.0)	40 (15.5)		
Space constraints	45 (14.3)	39 (15.1)		
Physical impairment	77 (24.5)	60 (23.3)		
Facilitators				
Perceived benefits	81 (30.5)	68 (30.1)	0.06	0.996
Family support	54 (20.3)	47 (20.8)		
Available space	64 (24.1)	54 (23.9)		
Motivation	67 (25.2)	57 (25.2)		

Note. Bold character is for significative associations.

**Table 5 ijerph-23-00051-t005:** Factors associated with physical activity after multivariable regression.

Variables	Crude OR (IC95%)	*p* Value	Adjusted OR (IC95%)	*p* Value
Age (years)				
20–29 (ref)	1.00	—	1.00	—
30–44	0.96 [0.55–1.65]	0.087	0.89 [0.80–0.99]	0.041
45–59	4.29 [2.50–7.40]	<0.001	3.12 [1.90–5.10]	<0.001
60–74	4.13 [2.40–7.10]	<0.001	2.84 [1.70–4.70]	<0.001
≥75	41.29 [15.0–110.1]	<0.001	18.67 [7.50–46.0]	<0.001
Sex				
Male (ref)	1.00	—	1.00	—
Female	1.64 [1.07–2.50]	0.023	1.42 [1.02–1.98]	0.038
Education level				
None (ref)	1.00	—	1.00	—
Primary	0.22 [0.12–0.40]	<0.001	0.31 [0.18–0.52]	<0.001
Secondary	0.08 [0.03–0.20]	<0.001	0.12 [0.06–0.25]	<0.001
High school	0.03 [0.01–0.08]	<0.001	0.05 [0.02–0.12]	<0.001
Comorbidities				
None (ref)	1.00	—	1.00	—
Renal failure	3.12 [1.90–5.10]	<0.001	2.41 [1.50–3.90]	<0.001
Diabetes/other	6.84 [3.80–12.0]	<0.001	4.92 [2.80–8.60]	<0.001

## Data Availability

The data that support the findings of this study are not publicly available due to privacy or ethical restrictions but are available from the corresponding author on reasonable request and pending editorial clearance.

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
