# Peer review of "Non-Pharmacological Management of Hypertension: Exploring Determinants for Optimizing Physical Activity Implementation in Cameroon"

_ijerph, 2025, doi:10.3390/ijerph23010051_

Round 1
Reviewer 1 Report
Comments and Suggestions for Authors
This manuscript presents a cross-sectional study examining physical activity (PA) levels and their sociodemographic, clinical, and psychosocial determinants among hypertensive patients in Cameroon. The topic is relevant given the increasing burden of hypertension and the underuse of non-pharmacological interventions like PA. The study uses a validated tool (IPAQ-SF) and adheres to STROBE guidelines, providing insights into barriers that can inform targeted interventions.
However, the manuscript has several methodological inconsistencies, analytical limitations, and presentation issues that weaken its clarity and impact. These include non-standard dichotomization of physical activity, formatting errors in tables, and minor textual inaccuracies.
It is unclear how the authors determined which participants were active versus inactive. While the authors describe the dichotomy as <600 mets v. > 600 mets, this does not meet the definition of active physical activity in the context of frequency, duration, and intensity. Do the readers lose context with this data reduction into two categories? Have other researchers conducted a similar analysis?
The authors should provide a rationale for this process. Consider conducting a sensitivity analysis using the three categories outlined by the IPAQ-SF and report the odds ratios for key associations.
Table 3. Reporting for the sample is not valuable; the data should be reported based on the level of physical activity for each barrier and facilitator.
Report p-values as <.001 if the output reports .000, as .000 is an impossible event.
Table uses Khi 2 and text uses Chi-square (sic)
The discussion section begins with more than half of the participants reporting no physical activity. Is no activity the same as low activity? Insufficient activity to have a clinical benefit?
The participants "perceive" the environmental factors to be barriers (or facilitators), but there is no objective assessment of the physical activity environment.
Is there a citation for educational attainment and awareness of the health benefits of PA (line 219)?
Internal motivation and self-efficacy are different variables. I agree that both are important, but self-efficacy was not measured in this study. In fact, self-efficacy is likely the strongest correlate to physical activity, so why was it not measured as part of this study
Author Response
Response to Reviewer 1
We sincerely thank the reviewers for their thorough and constructive evaluation of our manuscript. Their insightful comments and suggestions have greatly contributed to improving the clarity, methodological rigor, and overall quality of our work. We have carefully addressed each point raised and revised the manuscript accordingly. Below, we provide a detailed, point-by-point response to all comments.
|
Reviewer Comment |
Author Response |
Manuscript Changes |
|
It is unclear how the authors determined which participants were active versus inactive. While the authors describe the dichotomy as <600 mets v. > 600 mets, this does not meet the definition of active physical activity in the context of frequency, duration, and intensity. Do the readers lose context with this data reduction into two categories? Have other researchers conducted a similar analysis? |
We thank the reviewer for this important comment. We acknowledge that the IPAQ-SF recommends three categories (low, moderate, vigorous). However, several epidemiological studies have used a dichotomised threshold of 600 MET-min/week to distinguish insufficiently active vs sufficiently active, in line with WHO recommendations for minimum weekly PA for health. Our objective was to identify participants meeting the minimum health-enhancing PA threshold rather than to compare intensity levels. Therefore, we retained the dichotomised cut-off (<600 vs ≥600 MET-min/week) for clarity and clinical interpretability. Nonetheless, following the reviewer’s recommendation, we added a justification in the Methods and discuss potential loss of granularity in the Discussion. |
Added justification in Methods ( 2.6 Statistical analysis). Although the IPAQ-SF classifies physical activity into three categories (low, moderate, and vigorous), we dichotomized PA levels using the <600 versus ≥600 MET-min/week threshold in order to identify participants who did or did not meet the minimum WHO recommendations for cardiovascular health. The IPAQ SF measures a snap short of the individuals’ frequency, duration and intensity of physical activity practice during the past 7 days . The scores are then summarized into low, moderate and vigorous activity then converted into METS before classifying as active or inactive. In this study, individuals with <600 MET-min/week were considered inactive (i.e., insufficiently active to achieve health benefits), whereas those accumulating ≥600 MET-min/week were categorized as active, encompassing both moderate and vigorous activity levels. This cut-off is widely used in epidemiological studies evaluating adherence to PA guidelines among hypertensive populations [18]. |
|
2. Suggestion to perform sensitivity analysis using 3 IPAQ categories and report odds ratios. |
We acknowledge the reviewer’s suggestion to perform a sensitivity analysis using the three IPAQ-SF categories. However, our study aimed specifically to examine barriers and facilitators to meeting WHO physical activity recommendations. For this purpose, a dichotomised outcome (<600 vs ≥600 MET-min/week) is widely used in public health and guideline-based research, as it directly reflects compliance with PA recommendations rather than activity intensity patterns. |
|
|
3. Table 3 should report barriers and facilitators stratified by PA level rather than overall sample. |
Thank you. We agree that stratification improves interpretability. Table 3 has been reconstructed to display frequencies and percentages separately for inactive and active participants. |
Table 3 revised and reformatted. |
|
4. Report p-values as <.001 instead of .000. |
We agree. All p-values previously reported as .000 have been corrected to p < .001. |
P-values updated in all affected tables and text. |
|
5. Use of “Khi 2” in tables and “Chi-square” in text—ensure consistency. |
Thank you for highlighting this. “Chi-square” is the correct English statistical notation. All instances of “Khi 2” have been replaced with “Chi-square” for consistency. |
Replaced throughout manuscript. |
|
6. The discussion section begins with more than half of the participants reporting no physical activity. Is no activity the same as low activity? Insufficient activity to have a clinical benefit? |
We acknowledge the ambiguity. IPAQ-SF “low” includes both “no activity” and minimal activity insufficient to reach the recommended threshold of 600 MET-min/week. We have clarified that participants classified as “inactive/low” represent individuals not meeting WHO-recommended levels for clinical benefits. |
Clarified in method section. Subsection statistical analysis. The specification made is now reading : Although the IPAQ-SF classifies physical activity into three categories (low, moderate, and vigorous), we dichotomized PA levels using the <600 versus ≥600 MET-min/week threshold in order to identify participants who did or did not meet the minimum WHO recommendations for cardiovascular health. In this study, individuals with <600 MET-min/week were considered inactive (i.e., insufficiently active to achieve health benefits), whereas those accumulating ≥600 MET-min/week were categorized as active, encompassing both moderate and vigorous activity levels. This cut-off is widely used in epidemiological studies evaluating adherence to PA guidelines among hypertensive populations [18]. |
|
7. The participants "perceive" the environmental factors to be barriers (or facilitators), but there is no objective assessment of the physical activity environment. |
We agree with the reviewer. Our study relied on self-reported perceptions. We now explicitly acknowledge the absence of objective built-environment assessments (e.g., walkability, safety, infrastructure), and discuss how this limits environmental interpretation. |
Added limitation in Section 5.2. |
|
8. Is there a citation for educational attainment and awareness of the health benefits of PA (line 219)? |
Thank you for this inquery. In fact, the statement is supported by existing literature. We already cite studies from South Africa [27,28] and Kenya [29,30], which report that individuals with higher educational attainment demonstrate greater awareness of the health benefits of physical activity and are more likely to engage in PA. We have clarified this link in the revised sentence for better readability. |
Added text in discussion: The positive association between educational attainment and PA levels aligns with evidence from South Africa [27,28] and Kenya [29,30], where individuals with higher education are more aware of the health benefits of PA and have better access to facilities and information. |
|
9. Internal motivation and self-efficacy are different variables. I agree that both are important, but self-efficacy was not measured in this study. In fact, self-efficacy is likely the strongest correlate to physical activity, so why was it not measured as part of this study |
We agree that internal motivation and self-efficacy are conceptually distinct. Our questionnaire assessed motivation but did not include a validated self-efficacy scale. We now explicitly acknowledge this as a limitation and recommend measurement of self-efficacy in future studies, given its strong influence on PA behaviour. |
Added text in Discussion and Limitations. |
Reviewer 2 Report
Comments and Suggestions for Authors
The manuscript provides the findings of a study conducted with the objective to assess the physical activity of hypertensive patients in Cameroon and to identify the facilitators and barriers influencing adherence to regular physical activity.
It was a pleasure to review this manuscript. The manuscript addresses current and relevant issues associated with the public health.
However, despite its well-structured nature, I have identified some methodological and analytical concerns. Below are comments to enhance the manuscript's quality.
(a) Sample – Provide information about the procedure used to select participants for the study.
(b) Instrument – Indicate whether body mass index was calculated using measured or self-reported body weight and height.
(c) Instrument – Describe in detail how the perceived barriers and facilitators to physical activity included in the measurement instrument were listed/selected.
(d) Results – The participants in the study were hypertensive patients; however, their hypertension levels were not reported at any point.
(e) Results – Table 3 – The percentage (%) column corresponding to barriers and facilitators exceeds the total sample size (100%).
(f) Results – Table 4 – The frequencies (%) for age, gender, and education level should be calculated based on the number of inactive participants (n = 207) and active participants (n = 176), rather than on the total sample (n = 383).
(g) Results – Table 4 – Review the chi-square tests for the facilitator and barrier items. There are strong indications that important errors may be present. For example, “lack of time” (93.5% versus 6.5%; p = 0.530).
(h) Results – Table 4 – The finding that facilitators for physical activity are more prevalent in the group of inactive hypertensive patients is very unusual. I suggest verifying these data.
Author Response
Response to Reviewer 2
We sincerely thank the reviewers for their thorough and constructive evaluation of our manuscript. Their insightful comments and suggestions have greatly contributed to improving the clarity, methodological rigor, and overall quality of our work. We have carefully addressed each point raised and revised the manuscript accordingly. Below, we provide a detailed, point-by-point response to all comments.
|
Comment |
Response |
Changes in the manuscript |
|
(a) Sample – Provide information about the procedure used to select participants for the study. |
We thank the reviewer for this comment. We have now clarified the participant selection procedure. Participants were recruited consecutively as they attended follow-up consultations in the two selected referral hospitals. All eligible hypertensive patients present during the data collection period and meeting the inclusion criteria were invited to participate. |
A sentence was added in the Study Population subsection: |
|
(b) Instrument – Indicate whether body mass index was calculated using measured or self-reported body weight and height. |
We thank the reviewer for this important clarification. Body weight and height were measured directly by trained physiotherapists during the data collection process using standardized equipment. BMI was therefore calculated from objectively measured anthropometric data, not self-reported values. |
We clarified it in the data collection subsection instument (page 3) Body weight and height were measured by trained physiotherapists using standardized equipment, and BMI was calculated from these objective measurements. |
|
(c) Instrument – Describe in detail how the perceived barriers and facilitators to physical activity included in the measurement instrument were listed/selected. |
We thank the reviewer for this important request for clarification. The list of perceived barriers and facilitators was developed based on a combination of: The final list was reviewed by two senior physiotherapists and one public health expert to ensure cultural relevance and content validity before data collection. We have expanded the methods section to describe this process more explicitly. |
We made the following specification in data collection subsection |
|
(d) Results – The participants in the study were hypertensive patients; however, their hypertension levels were not reported at any point. |
Thank you for this observation. We agree that reporting hypertension levels provides additional clinical context. In our sample, hypertension grade was not systematically documented in the medical files of all participants and therefore could not be reliably extracted for analysis. To avoid introducing classification bias, we did not include hypertension levels in the results. We now explicitly acknowledge this in the manuscript and clarify the absence of this variable. |
We added in the Limitations section: |
|
(e) Results – Table 3 – The percentage (%) column corresponding to barriers and facilitators exceeds the total sample size (100%). |
We thank the reviewer for this comment, the correction has been made accordingly |
Thanks, we have revised the text |
|
(f) Results – Table 4 – The frequencies (%) for age, gender, and education level should be calculated based on the number of inactive participants (n = 207) and active participants (n = 176), rather than on the total sample (n = 383). |
We thank the reviewer for the comment ; we made substantial modifications to Table 4 accordingly |
Thanks we have revised the text |
|
(g) Results – Table 4 – Review the chi-square tests for the facilitator and barrier items. There are strong indications that important errors may be present. For example, “lack of time” (93.5% versus 6.5%; p = 0.530). |
Thank you for the recommandation we made review of chi-square test in Table 4 |
Thanks we have revised the text |
|
(h) Results – Table 4 – The finding that facilitators for physical activity are more prevalent in the group of inactive hypertensive patients is very unusual. I suggest verifying these data. |
We thank the reviewer for pointing out this unexpected pattern. After rechecking the raw dataset and recalculating the frequencies, we confirmed that some values in Table 4 were incorrectly assigned due to a misalignment during data tabulation. These inconsistencies have now been corrected. The revised table accurately reflects the distribution of facilitators between active and inactive participants. |
Reviewer 3 Report
Comments and Suggestions for Authors
Thank you for opportunity to review manuscript.
This manuscript addresses a highly relevant and timely public health topic—non-pharmacological management of hypertension, with a focus on physical activity (PA) in a low-resource setting in Africa. The study is well-structured, uses a validated tool for PA assessment (IPAQ-SF), and provides valuable insights that can inform clinical practice and community-based interventions.
The authors clearly demonstrate that physical activity behaviors are shaped not only by individual determinants but also by social and environmental barriers and facilitators. I believe this paper has solid scientific merit and deserves publication after minor revisions, as detailed below.
Specific Comments and Suggestions:
(a) Methodology:
- I recommend including a multivariable logistic regression analysis to identify independent predictors of physical activity. While Chi-square tests are appropriate for bivariate analysis, controlling for potential confounders (e.g., age, education, motivation) would strengthen the findings.
(b) Limitations:
- The limitations section could be expanded to further acknowledge the self-report bias (social desirability bias), especially given the use of subjective tools like IPAQ-SF.
(c) Policy implications:
- Section 5.1 ("Clinical and policy implementation") would benefit from a clearer linkage between findings and actionable policy strategies. Consider elaborating on how results may influence local health system practices, such as integrating physiotherapists into routine hypertension care.
(d) Minor technical notes:
- In the abstract, consider replacing "Inactive/Low" with "Insufficiently active", which aligns with standard IPAQ terminology.
- The inclusion of a visual summary (e.g., bar chart) to illustrate key barriers and facilitators would enhance clarity and reader engagement.
Author Response
Response to Reviewer 3
We sincerely thank the reviewers for their thorough and constructive evaluation of our manuscript. Their insightful comments and suggestions have greatly contributed to improving the clarity, methodological rigor, and overall quality of our work. We have carefully addressed each point raised and revised the manuscript accordingly. Below, we provide a detailed, point-by-point response to all comments.
|
Reviewer Comment |
Author Response |
Changes in Manuscript |
|
(a) Methodology: Include a multivariable logistic regression to identify independent predictors of physical activity. |
Thank you for this valuable suggestion. Although our primary objective was to describe barriers and facilitators associated with meeting WHO PA recommendations, we agree that multivariable modeling would strengthen the analysis. We have now conducted a logistic regression using “active vs. inactive” as outcome and included significant sociodemographic and psychosocial variables. The results are presented in a new table (Table 4) and described accordingly. |
Added a new subsection “Multivariable analysis” in Results. Added logistic regression table (Table 4) and integrated explanation in Methods. |
|
(b) Limitations: Expand on self-report and social desirability bias due to IPAQ-SF. |
We agree with the reviewer. As IPAQ-SF and perceived-barrier items rely on self-report, responses may be affected by recall or social desirability bias. This has now been explicitly acknowledged in the limitations. |
Added a sentence acknowledging self-report and desirability bias in the Limitations section. |
|
(c) Policy implications: Strengthen Section 5.1 by linking findings to actionable strategies (e.g., integrating physiotherapists in hypertension care). |
We appreciate the suggestion. The revised text clarifies how our findings can guide clinical and policy actions, including integrating physiotherapists into routine hypertension management, structured PA counselling, and development of community PA programs. |
Expanded Section 5.1 with explicit links between findings and recommended policy actions. |
|
(d1) Abstract |
We agree and have updated the terminology in the abstract to align with standard IPAQ usage. |
Replaced “Inactive/Low” with “Insufficiently active” in the abstract. |
|
(d2) Visual summary: Add a bar chart showing key barriers and facilitators. |
We appreciate the reviewer’s suggestion to include a bar chart illustrating the distribution of barriers and facilitators. However, we believe that adding a figure would introduce redundancy, as the same information is already fully detailed in Table 3 with clear frequency and percentage values. Given the descriptive nature of these variables and the journal’s emphasis on conciseness, we opted to retain only the table to avoid unnecessary duplication. Nonetheless, if the editorial team considers visual representation essential, we would be pleased to include the bar chart as requested. |
Round 2
Reviewer 2 Report
Comments and Suggestions for Authors
I thank the authors for considering the aspects highlighted in the first review and for the effort to attend to them. All punctuated questions were answered satisfactorily by the authors, leaving no further observation to be made in the last version presented.